# Evaluating the Competence of AI Chatbots in Answering Patient-Oriented Frequently Asked Questions on Orthognathic Surgery

**DOI:** 10.3390/healthcare13172114

**Published:** 2025-08-26

**Authors:** Ezgi Yüceer-Çetiner, Dilara Kazan, Mobin Nesiri, Selçuk Basa

**Affiliations:** 1Department of Oral and Maxillofacial Surgery, School of Dental Medicine, Bahçeşehir University, Istanbul 34357, Turkey; dilara.kazan@bau.edu.tr (D.K.); selcuk.basa@bau.edu.tr (S.B.); 2Department of Oral and Dental Health, Vocational School of Health Services, Bahçeşehir University, Istanbul 34353, Turkey; mobin.nesiri@bau.edu.tr

**Keywords:** artificial intelligence, clinical safety, health communication, orthognathic surgery, patient education

## Abstract

Objectives: This study aimed to evaluate the performance of three widely used artificial intelligence (AI) chatbots—ChatGPT-4, Gemini 2.5 Pro, and Claude Sonnet 4—in answering patient-oriented frequently asked questions (FAQs) related to orthognathic surgery. Given the increasing reliance on AI tools in healthcare, it is essential to evaluate their performance to provide accurate, empathetic, readable, and clinically appropriate information. Methods: Twenty FAQs in Turkish about orthognathic surgery were presented to each chatbot. The responses were evaluated by three oral and maxillofacial surgeons using a modified Global Quality Score (GQS), binary clinical appropriateness judgment, and a five-point empathy rating scale. The evaluation process was conducted in a double-blind manner. The Ateşman Readability Formula was applied to each response using an automated Python-based script. Comparative statistical analyses—including ANOVA, Kruskal–Wallis, and post hoc tests—were used to determine significant differences in performance among chatbots. Results: Gemini outperformed both GPT-4 and Claude in GQS, empathy, and clinical appropriateness (*p* < 0.001). GPT-4 demonstrated the highest readability scores (*p* < 0.001) but frequently lacked empathetic tone and safety-oriented guidance. Claude showed moderate performance, balancing ethical caution with limited linguistic clarity. A moderate positive correlation was found between empathy and perceived response quality (r = 0.454; *p* = 0.044). Conclusions: AI chatbots vary significantly in their ability to support surgical patient education. While GPT-4 offers superior readability, Gemini provides the most balanced and clinically reliable responses. These findings underscore the importance of context-specific chatbot selection and continuous clinical oversight to ensure safe and ethical AI-driven communication.

## 1. Introduction

Orthognathic surgery refers to a group of surgical procedures aimed at correcting dentofacial skeletal discrepancies. It is often indicated for patients with functional impairments such as malocclusion, speech difficulties, airway obstruction, and esthetic concerns. Through the combined correction of facial harmony and occlusal function, orthognathic surgery can lead to significant improvements in patients’ self-image, psychosocial well-being, and overall quality of life [1,2].

The decision to undergo this form of surgery is frequently accompanied by uncertainty and anxiety, prompting patients to seek detailed information about surgical indications, risks, recovery, and expected outcomes. When accurate and comprehensive information is not readily available, they may experience increased anxiety, reduced satisfaction with surgical outcomes, and difficulties in adhering to perioperative instructions [3]. While healthcare professionals continue to serve as the primary source of reliable patient counseling, the rise of digital health technologies—particularly AI-powered chatbots—has led many individuals to seek self-education and pre-consultation support online [4,5].

AI chatbots are language-based software tools that leverage artificial intelligence, machine learning, and natural language processing to interpret and respond to user queries [6]. With recent advancements in large language models, these systems have become significantly more accurate, accessible, and contextually aware. In recent years, AI-powered chatbots have been increasingly integrated into healthcare communication strategies, offering scalable and accessible support for patient education, symptom triage, and treatment guidance across various medical fields. As a result, AI chatbots are increasingly being integrated into healthcare communication, offering immediate and structured responses to health-related inquiries. This growing reliance on AI-generated information highlights the critical need to ensure that such responses are not only factually accurate but also readable, empathetic, and clinically appropriate [7,8,9].

While studies have evaluated chatbot performance in areas like orthodontics, endodontics, oncology, and general health education, few have explored their utility in surgical patient education [4,10,11,12,13,14]. This is especially relevant for orthognathic procedures, where outcomes may profoundly impact both physical function and facial aesthetics, and where patient expectations must be carefully managed through clear, compassionate communication. Inaccurate or overly simplistic information may lead to misconceptions, delay necessary treatment, or undermine the trust between patient and surgeon [2,3]. This study aims to evaluate the quality, readability, empathy, and clinical appropriateness of responses generated by leading AI chatbots—ChatGPT (OpenAI), Gemini (Google DeepMind), and Claude (Anthropic)—when answering patient-oriented frequently asked questions (FAQs) about orthognathic surgery.

## 2. Materials and Methods

### 2.1. Study Design

This study aimed to evaluate the performance of AI chatbots in answering frequently asked questions related to orthognathic surgery. As the analysis was conducted using publicly available online content without involving human subjects or personal data, ethical approval was not required.

### 2.2. Question Selection

Relevant patient questions were collected from social media and community platforms, including Facebook, Instagram, and X, as well as from publicly accessible health websites. A total of 20 frequently asked questions in Turkish, directly related to orthognathic surgery, were selected by an experienced oral and maxillofacial surgeon (S.B). The selected questions addressed key themes such as diagnosis, surgical planning, techniques, risks, and potential complications (Table 1). These questions reflect real-world patient queries expressed in layperson language rather than medically complex scenarios. To prevent possible bias, the oral and maxillofacial surgeon responsible for question selection was not involved in the chatbot evaluation process.

### 2.3. Chatbot Evaluation

We selected GPT-4 (OpenAI), Gemini 2.5 Pro (Google DeepMind), and Claude Sonnet 4 (Anthropic) for evaluation due to their widespread use, high language fluency, and status as the most advanced publicly available large language models (LLMs) as of 2025. These chatbots also represent different AI developers, ensuring architectural diversity in the comparison. To eliminate any contextual bias, each question was submitted in a new session with each chatbot, and all responses were recorded for analysis. The LLMs were not fine-tuned or supplemented with external data through retrieval-based methods; responses were generated using their native capabilities as publicly available, general-purpose models.

### 2.4. Evaluation Criteria

Responses generated by the chatbots were evaluated based on the following criteria:Global Quality Score (GQS): Three oral and maxillofacial surgeons (E.Y.Ç, D.K., and M.N) blinded to the chatbots, independently assessed each response using a 5-point Likert scale (1 = poor, 5 = excellent). The evaluation focused on the quality, accuracy, and comprehensiveness of the information provided. This tool was developed based on a modified version of the Global Quality Score [15].Clinical Appropriateness: Responses were evaluated to determine whether the chatbots promoted safe and medically responsible guidance. Each response was assessed using a binary (yes/no) question: “Does the chatbot appropriately recommend that the patient seek further evaluation and management from a qualified healthcare professional?” [4].Readability and Accessibility: To evaluate the readability and accessibility of chatbot responses generated in Turkish, the Ateşman Readability Formula was applied. This formula is specifically designed for the Turkish language and is based on the average sentence length and the average number of syllables per word, generating a score between 1 and 100. Higher scores indicate texts that are easier to read and understand, while lower scores suggest increased complexity [16,17]. All chatbot-generated answers were processed using a custom Python (v3.10.8; Python Software Foundation, USA) script that automated the calculation of Ateşman scores for each response. This quantitative assessment provided insight into how accessible and patient-friendly the language of each chatbot was, particularly in the context of health communication.Empathy Evaluation: Evaluators rated the chatbot responses for empathy and bedside manner using a five-point scale: 1 = not empathetic, 2 = slightly empathetic, 3 = moderately empathetic, 4 = empathetic, and 5 = very empathetic. Higher scores reflected a greater degree of empathy and a more patient-centered communication style.

### 2.5. Statistical Analysis

Data were analyzed using IBM SPSS Statistics version 23 (IBM Corp., Armonk, NY, USA). The normality of quantitative variables was assessed with the Shapiro–Wilk test. For variables that met the normality assumption across three or more independent groups, a one-way analysis of variance (ANOVA) was applied, with post hoc comparisons performed using the Tukey test. When the normality assumption was not met, the Kruskal–Wallis H test was used, and pairwise comparisons were conducted with the Dunn test. Associations between categorical variables were examined with the Fisher Exact test incorporating Monte Carlo correction, while multiple comparisons were evaluated using the Bonferroni-adjusted z test. Correlations between non-normally distributed quantitative variables were analyzed using Spearman’s rho. Inter-rater reliability was calculated with the intraclass correlation coefficient (ICC) and Fleiss’ kappa. Descriptive statistics for quantitative data are presented as mean ± standard deviation and median (minimum–maximum), whereas categorical data are expressed as frequency (n) and percentage (%). A two-sided *p*-value < 0.05 was considered statistically significant.

## 3. Results

The inter-rater agreement among the three evaluators was assessed using intraclass correlation coefficients (ICCs) and Fleiss’ kappa statistics. The ICC for the Global Quality Score (GQS) was 0.338 (95% CI: 0.178–0.502; *p* < 0.001), indicating weak agreement. For clinical appropriateness, Fleiss’ kappa was 0.931 (*p* < 0.001), reflecting very high agreement. The ICC for empathy evaluation was 0.641 (95% CI: 0.512–0.752; *p* < 0.001), indicating moderate agreement among evaluators (Table 2).

Statistical comparisons among the AI chatbots showed significant differences across all evaluated variables. The median GQS was 4.5 (range: 3–5) for Gemini, 4 (range: 3–5) for GPT, and 4 (range: 3–4) for Claude. The difference was statistically significant (*p* < 0.001). For empathy ratings, the median values were 5 (range: 4–5) for Gemini, 3.5 (range: 3–5) for GPT, and 4 (range: 2–5) for Claude, with a significant difference between the groups (*p* < 0.001). The mean Ateşman readability scores were 45.77 ± 7.51 for Gemini, 63.01 ± 9.97 for GPT, and 44.69 ± 13.42 for Claude. The difference was statistically significant (*p* < 0.001) (Table 3). Clinical appropriateness was evaluated separately by each of the three assessors. For Evaluator 1, 95% of Gemini’s responses, 20% of GPT’s responses, and 65% of Claude’s responses were considered appropriate (*p* < 0.001; Fisher’s Exact test with Monte Carlo correction). Similar findings were reported by Evaluator 2, with appropriateness rates of 95% for Gemini, 15% for GPT, and 65% for Claude (*p* < 0.001). Evaluator 3 recorded appropriateness rates of 95% for Gemini, 15% for GPT, and 70% for Claude (*p* < 0.001) (Table 4).

Spearman’s rho correlation analysis revealed that, for GPT, there was a moderate positive correlation between GQS and empathy scores (r = 0.454; *p* = 0.044). In the overall dataset, a moderate positive correlation was found between GQS and empathy (r = 0.443; *p* < 0.001), and a moderate negative correlation was found between GQS and Ateşman scores (r = −0.394; *p* = 0.002). No other statistically significant correlations were observed (*p* > 0.05) (Table 5).

## 4. Discussion

With the rapid advancement of artificial intelligence, AI-powered chatbots are increasingly being utilized in patient education, especially in clinical domains where the demand for information is high and access to healthcare professionals may be limited [10]. Orthognathic surgery is one such area, involving complex treatment planning and substantial functional and aesthetic outcomes that can significantly affect patients both physically and emotionally. In this context, clear, accurate, and empathetic communication is essential for helping patients form realistic expectations and make informed decisions [18,19].

This study expands the existing literature by evaluating the performance of three widely used AI chatbots—ChatGPT, Gemini, and Claude—in responding to frequently asked, patient-centered questions about orthognathic surgery. While previous research has assessed chatbot performance in fields such as general and dental health communication, surgical specialties pose distinct challenges owing to their clinical complexity and emotional impact [10,11,12,13,14]. These characteristics underscore the importance of evaluating not only informational accuracy but also empathy, readability, and clinical appropriateness. The findings presented here offer valuable insight into the capabilities and limitations of current AI chatbots in supporting patient education for surgical care.

The study’s methodology integrated both linguistic and clinical evaluation criteria to provide a comprehensive analysis of chatbot responses. Quality (using a modified Global Quality Score), clinical appropriateness, empathy, and readability were assessed to yield a multifaceted view of chatbot competence. Ateşman readability scores provide a standardized metric for comparing linguistic complexity and potential patient accessibility. The Ateşman Readability Formula—conceptually analogous to the Flesch Reading Ease and Flesch–Kincaid Grade Level formulas but adapted for Turkish—ensured that accessibility was judged with a culturally and linguistically relevant yardstick, which is an essential consideration in non-English-speaking settings where health literacy and the availability of comprehensible materials remain limited [16,20]. Furthermore, the inclusion of an empathy metric added an important dimension to the evaluation. Prior work indicates that tone and simulated bedside manner, even when delivered by AI, shape patients’ perceptions and trust. By incorporating empathy scoring, the present study acknowledges that effective health communication depends not only on factual accuracy but also on emotional resonance and delivery style [21].

Marked performance differences emerged among the three chatbots, reflecting contrasts not only in underlying architectures but also in design philosophies and intended user experiences. Gemini 2.5 Pro consistently outperformed GPT-4 and Claude 4 Sonnet across most criteria, achieving the highest scores for global response quality, empathy, and clinical appropriateness. This superiority aligns with Gemini’s reinforcement-learning approach, which prioritizes contextual awareness and safety through alignment with real-world data [22,23]. By contrast, GPT-4 produced the most readable responses but scored markedly lower in empathy and clinical appropriateness—a trade-off illustrating that simplified language does not automatically ensure adequate safety guidance or emotional sensitivity [24]. Claude performed moderately in all categories, displaying more empathy and clinical caution than GPT-4 but lacking the clarity and detail seen in Gemini’s outputs. Its Constitutional-AI-driven conservatism may minimize misinformation risk yet yield somewhat vague communication [25]. Given the absence of previous studies directly comparing these specific AI chatbots in the context of orthognathic surgery, direct comparisons with the existing literature were not possible, and interpretations were necessarily contextual.

Correlations among quality, empathy, and readability reinforced the nuanced trade-offs inherent in health-related chatbot communication. Emotionally resonant answers tended to be judged more informative, whereas higher readability often coincided with reduced clinical depth. A further observation concerned the frequency of safety-oriented recommendations: Gemini and Claude regularly advised professional consultation, whereas GPT-4 seldom did so, highlighting divergent model objectives and alignment strategies. To assess the reliability of these evaluations, inter-rater agreement was calculated using intraclass correlation coefficients and Fleiss’ kappa, which demonstrated moderate to high consistency, particularly for clinical appropriateness and empathy scoring. These results strengthen the credibility of the comparative assessments presented. However, the relatively low ICC observed for the quality score may reflect inherent subjectivity in evaluating the comprehensiveness and accuracy of AI-generated content, even among calibrated reviewers.

Several limitations warrant consideration. First, the evaluation encompassed only 20 Turkish-language questions, which may not capture the full spectrum of patient concerns. While our selection aimed to reflect the most common and clinically relevant issues, this limited sample may not fully represent the diversity of patient perspectives and is acknowledged as a limitation of the study. Second, AI chatbot models are frequently updated, and their behavior can shift rapidly; the responses analyzed here, therefore, represent a single point in time. Longitudinal studies are needed to monitor the consistency and evolution of chatbot performance across successive model versions. Third, although evaluators were blinded to chatbot identities and employed structured rating tools, the inherently subjective nature of quality and empathy scoring may introduce variability. Fourth, this study assessed only text-based outputs; voice delivery, multimedia explanations, and adaptive follow-up functions—features likely to influence real-world engagement—were outside its scope. Also, structural and accessibility-related differences among the chatbots may have influenced their performance. Factors such as whether the models are free or subscription-based, the number of countries in which they are widely used, and the duration of Turkish language support could have affected the extent and quality of their training in Turkish-language medical communication. These aspects warrant consideration when interpreting the comparative results. Additionally, although obtaining questions directly from individuals who have undergone orthognathic surgery could have enriched the dataset with firsthand perspectives, this method would have required ethical approval and patient recruitment, which were beyond the current study’s scope. This limitation is acknowledged, and future studies incorporating direct patient input are encouraged. Moreover, as chatbot technologies evolve, integrating visual tools such as facial simulation or morphing-based estimations of post-surgical outcomes could further enhance patient understanding and support shared decision-making in orthognathic surgery.

In light of these findings, several practical recommendations may guide the integration of chatbot responses into clinical settings. First, chatbot outputs should be used as adjunct tools to support, not replace, professional consultation. Embedding them within official patient portals or preoperative educational platforms can enhance access while maintaining oversight. Furthermore, regular clinical review of chatbot-generated content, as well as fine-tuning models with domain-specific communication datasets, may improve their alignment with patient needs and safety standards.

Despite these constraints, the findings underline the potential utility of AI chatbots in surgical patient education. In the context of Turkey, orthognathic patients may encounter long waiting lists and limited access to multidisciplinary care in public hospitals, while private sector services may offer faster treatment but often with limited preoperative counseling. These disparities can increase patients’ reliance on online sources for preliminary information. Therefore, the integration of AI-based chatbots into digital health communication may help bridge this informational gap, particularly by providing immediate, consistent, and accessible answers to frequently asked questions, regardless of the healthcare setting. In clinical practice, such tools may reinforce perioperative instructions, clarify complex procedures, and provide emotional support. Nevertheless, integration into patient-facing workflows should proceed with caution. Given the variability observed in accuracy, empathy, and safety content, healthcare providers must critically appraise chatbot outputs before dissemination. Future research should explore chatbot performance across diverse surgical specialties, languages, and patient populations, ideally incorporating dynamic interaction scenarios such as follow-up questioning or individualized advice. Monitoring models over time will be crucial for capturing performance drift or improvement. Ultimately, collaboration among clinicians, AI developers, and health-communication experts will be vital to produce chatbots that are both technologically advanced and ethically sound.

## 5. Conclusions

AI chatbots show promise as supplementary tools for surgical patient education, yet their effectiveness varies markedly by model. GPT-4 offers superior readability but may lag in clinical caution and empathy. Gemini delivers the best overall balance of accuracy, safety, and emotional tone, whereas Claude provides a conservative, ethically focused alternative at the cost of detail and clarity. These findings highlight the importance of model selection tailored to clinical intent and reinforce the need for ongoing clinician oversight to ensure safe, responsible, and effective AI-driven health communication.

## Figures and Tables

**Table 1 healthcare-13-02114-t001:** Frequently asked questions on orthognathic surgery.

What is orthognathic surgery? Who is it suitable for? Is jaw surgery absolutely necessary, or is orthodontic treatment alone sufficient? Is the upper jaw, lower jaw, or both operated on in orthognathic surgery? Is it mandatory to get braces before jaw surgery? Is orthodontic treatment required after jaw surgery? When can I return to work or school after orthognathic surgery? How long does orthognathic surgery take? Will there be visible scars on my face after jaw surgery? How long is the recovery period after orthognathic surgery? Is there pain after orthognathic surgery? Will there be swelling and bruising on my face after orthognathic surgery? What kind of diet should I follow after orthognathic surgery? Is numbness common after orthognathic surgery? Will my speech be affected after orthognathic surgery? How will my appearance change after orthognathic surgery? What should I pay attention to after orthognathic surgery? What are the risks of orthognathic surgery? How are the bones fixed in orthognathic surgery? Do the plates and screws remain for life? How often are follow-up appointments scheduled after orthognathic surgery? Is it possible that a second surgery will be needed after orthognathic surgery?

**Table 2 healthcare-13-02114-t002:** Inter-rater agreement analysis among evaluators.

	ICC (%95 CI)/Fleiss’ Kappa	*p*
Global Quality Score	0.338 (0.178–0.502)	<0.001 ^x^
Clinical Appropriateness	0.931	<0.001 ^y^
Empathy Evaluation	0.641 (0.512–0.752)	<0.001 ^x^

^x^ ICC: Intraclass correlation coefficient; ^y^ Fleiss’ kappa.

**Table 3 healthcare-13-02114-t003:** Comparison of variables by AI tools.

	Gemini 2.5 Pro	GPT-4	Claude Sonnet 4	Test Statistic	*p*
Global Quality Score	4.5 (3–5) ^a^	4 (3–5) ^b^	4 (3–4) ^b^	16.638	<0.001 ^x^
Empathy Evaluation	5 (4–5) ^a^	3.5 (3–5) ^b^	4 (2–5) ^b^	27.949	<0.001 ^x^
Ateşman Scores	45.77 ± 7.51 ^b^	63.01 ± 9.97 ^a^	44.69 ± 13.42 ^b^	18.871	<0.001 ^y^

^x^ Kruskal–Wallis H test; ^y^ one-way analysis of variance; mean ± standard deviation and median (minimum–maximum); ^a,b^: no significant difference between AI tools sharing the same letter.

**Table 4 healthcare-13-02114-t004:** Clinical appropriateness across AI tools.

	Gemini 2.5 Pro	GPT-4	Claude Sonnet 4	Total	Test Statistic	*p*
Clinical Appropriateness-1						
No	1 (5) ^a^	16 (80) ^b^	7 (35) ^a^	24 (40)	25.054	<0.001 ^x^
Yes	19 (95) ^a^	4 (20) ^b^	13 (65) ^a^	36 (60)		
Clinical Appropriateness-2						
No	1 (5) ^a^	17 (85) ^b^	7 (35) ^a^	25 (41.7)	28.651	<0.001 ^x^
Yes	19 (95) ^a^	3 (15) ^b^	13 (65) ^a^	35 (58.3)		
Clinical Appropriateness-3						
No	1 (5) ^a^	17 (85) ^b^	6 (30) ^a^	24 (40)	29.334	<0.001 ^x^
Yes	19 (95) ^a^	3 (15) ^b^	14 (70) ^a^	36 (60)		

^x^ Fisher’s Exact test with Monte Carlo correction; n (%); ^a,b^: no significant difference between AI tools sharing the same letter.

**Table 5 healthcare-13-02114-t005:** Intervariable analysis of AI tools.

		Global Quality Score	
		r	*p*
Gemini 2.5 Pro	Empathy Evaluation	−0.014	0.954
	Ateşman Scores	0.255	0.277
GPT-4	Empathy Evaluation	0.454	0.044
	Ateşman Scores	0.058	0.810
Claude Sonnet 4	Empathy Evaluation	−0.140	0.556
	Ateşman Scores	0.150	0.527
Overall	Empathy Evaluation	0.443	<0.001
	Ateşman Scores	−0.394	0.002

Spearman’s rho correlation coefficient.

## Data Availability

The raw data supporting the conclusions of this article will be made available by the authors on request.

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
