# Peer review of "Evaluating the Competence of AI Chatbots in Answering Patient-Oriented Frequently Asked Questions on Orthognathic Surgery"

_healthcare, 2025, doi:10.3390/healthcare13172114_

Round 1
Reviewer 1 Report
Comments and Suggestions for Authors
Dear Authors
I would like congratulate you for conducting such an interesting study.
I have the following questions:
1- Why did you choose these AI chatbots?
2- Did you think that you cover all FAQs?
3- Did you think that collecting questions from those that underwent orthognathic surgery will be beneficial?
Regards
Author Response
We thank the reviewer for these valuable comments, which have helped improve the clarity and scientific rigor of the manuscript.
Comment 1: Why did you choose these AI chatbots?
Response 1: Thank you for your question. We agree with this comment. We selected GPT-4 (OpenAI), Gemini 2.5 Pro (Google DeepMind), and Claude Sonnet 4 (Anthropic) based on their status as the most advanced and widely recognized large language models (LLMs) available to the public at the time of data collection. These three systems represent leading AI chatbot technologies developed by three different major companies, ensuring diversity in architecture and training methodologies. Their popularity and documented use in prior healthcare-related evaluations also make them relevant for comparative analysis. This approach allowed us to benchmark chatbot performance across multiple state-of-the-art platforms rather than focusing on a single system.
Therefore we have added more rationale for chatbot selection to the Materials and Methods / Chatbot Evaluation section (Page 3, Paragraph 2, Line 91) in highlight as:
"We selected GPT-4 (OpenAI), Gemini 2.5 Pro (Google DeepMind), and Claude Sonnet 4 (Anthropic) for evaluation due to their widespread use, high language fluency, and status as the most advanced publicly available large language models (LLMs) as of 2025. These chatbots also represent different AI developers, ensuring architectural diversity in the comparison."
Comment 2: Did you think that you cover all FAQs?
Response 2: Thank you for pointing this out. We acknowledge that it is not possible to cover every potential patient question in a single study. However, we aimed to include a representative and comprehensive set of FAQs that would reflect the most common and clinically relevant topics in orthognathic surgery. To achieve this, we collected a wide range of questions from publicly available sources, including social media platforms and reputable health websites, and selected the 20 most frequently encountered questions after careful review by an experienced oral and maxillofacial surgeon. These questions spanned key domains such as diagnosis, surgical planning, procedure details, recovery, and complications, which are commonly addressed in preoperative consultations.
So, a statement regarding this limitation and representativeness has been added to the Discussion section (Page 7, Paragraph 3, Line 239) in highlight as:
"Several limitations warrant consideration. First, the evaluation encompassed only 20 Turkish-language questions, which may not capture the full spectrum of patient concerns. While our selection aimed to reflect the most common and clinically relevant issues, this limited sample may not fully represent the diversity of patient perspectives and is acknowledged as a limitation of the study."
Comment 3: Did you think that collecting questions from those that underwent orthognathic surgery will be beneficial?
Response 3: Yes, we agree that collecting questions directly from patients with prior orthognathic surgery experience could further enrich the dataset. However, this approach would require ethical approval and patient recruitment, which was beyond the scope of this study. We have acknowledged this as a limitation and recommended it for future research.
We added a paragraph regarding the Discussion section (Page 7, Paragraph 3, Line 254) in highlight as:
"Additionally, although obtaining questions directly from individuals who have undergone orthognathic surgery could have enriched the dataset with firsthand perspectives, this method would have required ethical approval and patient recruitment, which were beyond the current study’s scope. This limitation is acknowledged, and future studies incorporating direct patient input are encouraged."
Reviewer 2 Report
Comments and Suggestions for Authors
- The idea of this research work is original, and the study design is ok.
- Statistical analyses are appropriately applied.
- I am concerned about the sample size of 20 FAQs; it may not fully represent the range of patient concerns. The authors should justify this limitation.
- The ICC for GQS was low (0.338), which suggests a potential inconsistency in quality assessments. The authors should provide possible reasons for this.
- Kindly add practical recommendations for the implementation of chatbot outputs.
- The Keywords for instance health communication and clinical safety may be added.
- Overall, the study adds valuable contributions to scientific literature and may be considered for publication after suggested revisions.
Author Response
Comment 1: The idea of this research work is original, and the study design is ok.
Response 1: We sincerely thank the reviewer for recognizing the originality of our study and for the positive feedback regarding the study design.
Comment 2: Statistical analyses are appropriately applied.
Response 2: We appreciate the reviewer’s upbeat assessment of the statistical analysis applied in our study.
Comment 3: I am concerned about the sample size of 20 FAQs; it may not fully represent the range of patient concerns. The authors should justify this limitation.
Response 3: Thank you for pointing this out. We acknowledge that it is not possible to cover every potential patient question in a single study. However, we aimed to include a representative and comprehensive set of FAQs that would reflect the most common and clinically relevant topics in orthognathic surgery. To achieve this, we collected a wide range of questions from publicly available sources, including social media platforms and reputable health websites, and selected the 20 most frequently encountered questions after careful review by an experienced oral and maxillofacial surgeon. These questions spanned key domains such as diagnosis, surgical planning, procedure details, recovery, and complications, which are commonly addressed in preoperative consultations.
So, a statement regarding this limitation and representativeness has been added to the Discussion section (Page 7, Paragraph 3, Line 239) in highlight as:
"Several limitations warrant consideration. First, the evaluation encompassed only 20 Turkish-language questions, which may not capture the full spectrum of patient concerns. While our selection aimed to reflect the most common and clinically relevant issues, this limited sample may not fully represent the diversity of patient perspectives and is acknowledged as a limitation of the study."
Comment 4: We appreciate the reviewer’s observation. The relatively low ICC value for GQS (0.338) likely reflects the subjective nature of quality assessments in evaluating AI-generated content, particularly when considering nuances such as medical accuracy, completeness, and clarity. Although all reviewers were calibrated before scoring, differences in interpretation may still arise due to varied clinical experiences and expectations regarding chatbot performance. This variation underscores the inherent challenge of standardizing human judgment in emerging areas like AI-generated health communication.
Therefore, we have added this explanation to the Discussion section (Page 7, Paragraph 2, Line 233) in highlight as:
"However, the relatively low ICC observed for the quality score may reflect inherent subjectivity in evaluating the comprehensiveness and accuracy of AI-generated content, even among calibrated reviewers."
Comment 5: Kindly add practical recommendations for the implementation of chatbot outputs.
Response 5: Thank you for this helpful suggestion. We agree that offering guidance on the clinical integration of chatbot outputs enhances the applicability of our findings.
Therefore, we have added a new paragraph to the Discussion section in highlight (Page 7, Paragraph 4, Line 263) as:
"In light of these findings, several practical recommendations may guide the integration of chatbot responses into clinical settings. First, chatbot outputs should be used as adjunct tools to support, not replace, professional consultation. Embedding them within official patient portals or preoperative educational platforms can enhance access while maintaining oversight. Furthermore, regular clinical review of chatbot-generated content, as well as fine-tuning models with domain-specific communication datasets, may improve their alignment with patient needs and safety standards."
Comment 6: The Keywords for instance health communication and clinical safety may be added.
Response 6: Thank you for your suggestion. We agree that the inclusion of “health communication” and “clinical safety” would enhance the discoverability and relevance of the manuscript. We have added both terms to the list of Keywords (Page 1) in the revised version of the manuscript in highlight.
Comment 7: Overall, the study adds valuable contributions to scientific literature and may be considered for publication after suggested revisions.
Response 7: We sincerely appreciate your positive assessment of our work and your constructive suggestions. Your feedback has been instrumental in refining the manuscript, and we have addressed all points accordingly in the revised version.
Comment 8: The English could be improved to more clearly express the research.
Response 8: Thank you for your observation. We have thoroughly reviewed the manuscript to enhance clarity, improve sentence structure, and ensure smoother flow throughout the text. Grammar and phrasing have been revised where necessary to improve overall readability and precision of expression. We hope these improvements make the research easier to follow for an international audience.
Comment 9: Does the introduction provide sufficient background and include all relevant references? — Can be improved.
Response 9: We appreciate the reviewer’s constructive suggestion. In response, we revised the Introduction to improve its clarity, fluency, and contextual relevance. The opening sentence was updated to more accurately define orthognathic surgery as a group of procedures rather than a single operation. To better support the rationale of the study, we also expanded the final part of the Introduction with additional context on the growing use of AI-powered chatbots in healthcare communication. These revisions aim to provide a stronger foundation for the research without altering the original reference framework. All changes are implied as highlighted.
"Orthognathic surgery refers to a group of surgical procedures aimed at correcting dentofacial skeletal discrepancies." (Page 1, Paragraph 1, Line 38) "In recent years, AI-powered chatbots have been increasingly integrated into healthcare communication strategies, offering scalable and accessible support for patient education, symptom triage, and treatment guidance across various medical fields." (Page 2, Paragraph 2, Line 55)
Comment 10: Are the methods adequately described? — “Can be improved”
Response 10: Thank you for pointing this out. To follow up on your suggestion, we have revised the Materials and Methods Section/Chatbot Evaluation for more clarification. We implemented the changes as highlighted.
"We selected GPT-4 (OpenAI), Gemini 2.5 Pro (Google DeepMind), and Claude Sonnet 4 (Anthropic) for evaluation due to their widespread use, high language fluency, and status as the most advanced publicly available large language models (LLMs) as of 2025. These chatbots also represent different AI developers, ensuring architectural diversity in the comparison." (Page 3, Paragraph 2, Line 91).
Reviewer 3 Report
Comments and Suggestions for Authors
Thank you for the opportunity to review this interesting article. I believe it raises several important questions. In my opinion, patients undergoing orthognathic surgery often have some unanswered concerns. Therefore, it is critical to explore the potential use of tools such as those investigated in this study, particularly as they are used almost daily by younger generations—who constitute a significant portion of orthognathic patients. The research is well designed and clearly written, with sound statistical analysis. However, I have several concerns and suggestions. I recommend revising the first sentence of the introduction to reflect that orthognathic surgery constitutes a group of procedures or a field of surgery, rather than a single operation. It would be beneficial to discuss in greater detail how this tool is applicable specifically within the context of Turkey. I suggest elaborating on the challenges that orthognathic patients may face in Turkey. Additionally, consider addressing whether there are differences in these issues between the private sector and public hospitals. It may also be interesting to discuss whether chatbots that patients can use might have a role in providing visualizations of expected final outcomes. I have some concerns about the study design. I am curious whether the three selected chatbots are indeed the most widely used in Turkey. If they are, please acknowledge this; if not, please explain the rationale for their selection. Thank you again for the opportunity to review this interesting article.
Author Response
We sincerely thank the reviewer for their thoughtful and constructive feedback. We appreciate your recognition of the study’s relevance, clarity, and design. Your comments have helped us further refine the manuscript, and we have carefully addressed each suggestion to improve the overall quality and contextual depth of the work.
Comment 1: I recommend revising the first sentence of the introduction to reflect that orthognathic surgery constitutes a group of procedures or a field of surgery, rather than a single operation.
Response 1: Thank you for your valuable observation. We have revised the first sentence of the Introduction to clarify that orthognathic surgery represents a field of surgery encompassing a group of procedures. We implemented the changes as highlighted.
"Orthognathic surgery refers to a group of surgical procedures aimed at correcting dentofacial skeletal discrepancies." (Page 1, Paragraph 1, Line 38)
Comment 2: It would be beneficial to discuss in greater detail how this tool is applicable specifically within the context of Turkey. I suggest elaborating on the challenges that orthognathic patients may face in Turkey. Additionally, consider addressing whether there are differences in these issues between the private sector and public hospitals.
Response 2: Thank you for this insightful suggestion. Specifically, we addressed the challenges related to long waiting times in public hospitals and the variability in access to multidisciplinary orthognathic care. We also acknowledged the increasing reliance on digital information sources, particularly in the private sector, where direct consultation may occur later in the decision-making process. These additions emphasize how AI chatbot tools could bridge the information gap for Turkish patients navigating different healthcare systems.
We have added a few sentences to the Discussion section, as highlighted, to reflect the Turkish healthcare context more explicitly.
" In the context of Turkey, orthognathic patients may encounter long waiting lists and limited access to multidisciplinary care in public hospitals, while private sector services may offer faster treatment but often with limited preoperative counseling. These disparities can increase patients’ reliance on online sources for preliminary information. Therefore, the integration of AI-based chatbots into digital health communication may help bridge this informational gap, particularly by providing immediate, consistent, and accessible answers to frequently asked questions, regardless of the healthcare setting." (Page 8, Paragraph 1, Line 271)
Comment 3: It may also be interesting to discuss whether chatbots that patients can use might have a role in providing visualizations of expected final outcomes.
Response 3: We appreciate the reviewer’s valuable suggestion. A sentence has been added to the Discussion section to highlight the potential of chatbots in delivering visual simulations of expected outcomes, which could enhance patient understanding and decision-making in orthognathic surgery. "Moreover, as chatbot technologies evolve, integrating visual tools such as facial simulation or morphing-based estimations of post-surgical outcomes could further enhance patient understanding and support shared decision-making in orthognathic surgery." (Page 7, Paragraph 3, Line 259, highlighted in green)
Comment 4: I have some concerns about the study design. I am curious whether the three selected chatbots are indeed the most widely used in Turkey. If they are, please acknowledge this; if not, please explain the rationale for their selection.
Response 4: Thank you for your evaluation. We agree with this comment. We selected GPT-4 (OpenAI), Gemini 2.5 Pro (Google DeepMind), and Claude Sonnet 4 (Anthropic) based on their status as the most advanced and widely recognized large language models (LLMs) available to the public at the time of data collection. These three systems represent leading AI chatbot technologies developed by three different major companies, ensuring diversity in architecture and training methodologies. Their popularity and documented use in prior healthcare-related evaluations also make them relevant for comparative analysis. This approach allowed us to benchmark chatbot performance across multiple state-of-the-art platforms rather than focusing on a single system.
Therefore, we have added more rationale for chatbot selection to the Materials and Methods / Chatbot Evaluation section (Page 3, Paragraph 2, Line 91) in highlight as:
"We selected GPT-4 (OpenAI), Gemini 2.5 Pro (Google DeepMind), and Claude Sonnet 4 (Anthropic) for evaluation due to their widespread use, high language fluency, and status as the most advanced publicly available large language models (LLMs) as of 2025. These chatbots also represent different AI developers, ensuring architectural diversity in the comparison."
Comment 5: Is the research design appropriate?” → Can be improved
Response 5: We appreciate the reviewer’s comment regarding the research design. While the current study design was intentionally developed to simulate real-world chatbot usage without altering model outputs or interaction structure, we acknowledge its inherent limitations. Specifically, we recognize that evaluating only text-based, one-time responses may not capture the full potential of interactive chatbot functionality. These limitations have been discussed in the Discussion section, where we have acknowledged the constraints of our design and emphasized the need for future studies incorporating dynamic, user-specific interactions and longitudinal assessments.
Comment 6: Are the methods adequately described? → Can be improved
Response 6: Thank you for pointing this out. To follow up on your suggestion, we have revised the Materials and Methods Section/Chatbot Evaluation for more clarification. We implemented the changes as highlighted.
"We selected GPT-4 (OpenAI), Gemini 2.5 Pro (Google DeepMind), and Claude Sonnet 4 (Anthropic) for evaluation due to their widespread use, high language fluency, and status as the most advanced publicly available large language models (LLMs) as of 2025. These chatbots also represent different AI developers, ensuring architectural diversity in the comparison." (Page 3, Paragraph 2, Line 91).
Round 2
Reviewer 3 Report
Comments and Suggestions for Authors
Authors managed to succesfully answer my concers and therefore improve their article.
Author Response
We sincerely thank the reviewer for their positive feedback and appreciation. We are pleased to hear that our revisions have successfully addressed the concerns and contributed to improving the quality of the manuscript.